# Transcription Factor Deformed Wings Is an Atg8a-Interacting Protein That Regulates Autophagy

**DOI:** 10.3390/cells13221897

**Published:** 2024-11-17

**Authors:** Marta Kołodziej, Panagiotis Tsapras, Alexander D. Cameron, Ioannis P. Nezis

**Affiliations:** School of Life Sciences, University of Warwick, Coventry CV4 7AL, UK; marta.geborys@warwick.ac.uk (M.K.); panos.tsapras.1@warwick.ac.uk (P.T.); a.cameron@warwick.ac.uk (A.D.C.)

**Keywords:** autophagy, LIR motif, transcription factors

## Abstract

LC3 (microtubule-associated protein 1 light chain 3, called Atg8 in yeast and *Drosophila*) is one of the most well-studied autophagy-related proteins. LC3 controls the selectivity of autophagic degradation by interacting with LIR (LC3-interacting region) motifs also known as AIM (Atg8-interacting motifs) on selective autophagy receptors that carry cargo for degradation. Although the function of Atg8 family proteins is primarily cytoplasmic, they are also enriched in the nucleus. Despite the accumulating evidence indicating the presence of Atg8 proteins in the nucleus, the mechanisms by which they are targeted to the nucleus, their interactions with nuclear components, and their nuclear role in remain poorly understood. Here, we used yeast two-hybrid screening, and we identified transcription factor Deformed wings (Dwg) as an Atg8a-interacting protein in *Drosophila*. Dwg-Atg8a interaction is LIR motif-dependent. We have created Dwg Y129A/I132A LIR mutant flies and shown that they exhibit elevated autophagy, improved resistance to oxidative stress, and starvation. Our results provide novel insights into the transcriptional regulation of autophagy in *Drosophila*.

## 1. Introduction

Autophagy is an evolutionarily conserved lysosomal degradation process involved in the breakdown and recycling of intracellular components [1,2]. There are three main types of autophagy: macroautophagy, microautophagy, and chaperone-mediated autophagy, all of which rely on the lysosome to digest intercellular cargo [3]. During macroautophagy, a double-membrane phagophore is formed to surround and isolate cargo. The enclosed cargo is then transported to the lysosome where it is broken down by hydrolytic enzymes and its components are returned to the cytosol to be re-utilized by the cell [1,2]. Macroautophagy is the most-studied type of autophagy and will hereafter be referred to as simply autophagy. Upregulation in the autophagy pathway is typically a survival response to nutrient depravation, although autophagy is constantly active at basal levels in most cell types, allowing for general housekeeping functions and the protection of intracellular integrity [1]. Autophagy also plays an important role in the quality control of proteins and organelles, making it a crucial process in the maintenance of homeostasis, the dysfunction of which can lead to abnormal protein build up and many pathologies [4].

Autophagy was considered to be regulated exclusively by cytosolic processes; however, increasing evidence over the years has shown that transcriptional and epigenetic events are crucial to autophagy regulation. Although not much is currently known about the transcriptional regulation of autophagy-related (ATG) genes, several transcription factors have been suggested to activate or suppress the expression of ATG genes [5,6]. Following the discovery that transcription factor EB (TFEB) can regulate a wide range of autophagy-related genes in response to nutrient deprivation, research into the transcriptional regulation of autophagy has significantly increased, leading to the discovery of additional transcription factors directly involved in autophagy regulation [7,8,9]. Transcriptional dysregulation in turn is often the cause of many human syndromes and complex diseases, including autoimmunity, neurological conditions, and developmental disorders, frequently due to signaling pathways targeting transcription machinery linked to autophagy regulation [6].

Microtubule-associated protein 1 light chain 3 (LC3; known as Atg8 in *Drosophila*) is one of the most-studied autophagy-related proteins. Among other ATG proteins, LC3 is essential in autophagy and is required for the elongation and maturation of the autophagosome. In recent years, the LC3-interacting region (LIR) or Atg8-interacting motif (AIM) has also been discovered to mediate interaction with Atg8 family proteins, providing invaluable insight into the function of selective autophagy [10]. Selective autophagy receptors (SARs) allow cargo to be degraded by interacting with Atg8 family proteins like LC3 via short amino acid LIR sequences, which bind to the Atg8 proteins via the LIR docking site (LDS), inducing autophagosome formation. Many ATG8/LC3-interacting proteins contain an LIR motif with a core sequence of (W/F/Y) XX(L/I/V) [11].

By utilizing high-throughput yeast two-hybrid (Y2H) screening, we have identified the transcription factor Deformed wings (Dwg) as a novel Atg8a-interacting protein, and we show that Dwg binds Atg8a in an LIR motif-dependent manner. We also demonstrate that Dwg colocalizes with Atg8a in both fed and starved conditions. To further characterize the significance of this interaction, we have used CRISPR technology and created Dwg Y129A/I132A LIR motif mutant flies. The Dwg Y129A/I132A LIR motif mutant flies show increased Atg8a lipidation and increased resistance to starvation and oxidative stress.

## 2. Results and Discussion

### 2.1. Dwg Is an Atg8a-Interacting Protein

To identify novel Atg8a-interacting proteins in *Drosophila*, we performed Y2H screening using *Drosophila* Atg8a (1–121), as a LexA bait (pB27) and an inducible LexA bait fusion (pB31), performed on a *Drosophila* 3rd instar larvae library. One of the positive hits is the transcription factor Dwg (also known as Zw5) (Figure 1A). Dwg plays a role in chromatin organization and gene regulation [12]. It consists of several zinc finger domains, including those of the C2H2 type, which facilitate DNA binding (Figure 1A). Y2H analysis revealed that Dwg interacts with Atg8a via the domain that corresponds to amino acids 1–180. (Figure 1A).

We confirmed the direct interaction between Dwg and Atg8a using glutathione S- transferase (GST) pull-down binding assays (Figure 1B). This interaction was significantly reduced when we used a mutant of Atg8a in which the LIR motif docking site (LDS) (Y49A) was impaired, indicating that the interaction between Dwg and Atg8a is LIR motif-dependent (Figure 1B). Dwg’s 1–180 domain contains a putative LIR motif YVMI at position 129–132 as identified by iLIR software version 1 [13]. This LIR motif contains a tyrosine residue at position +0 and an isoleucine residue at position +3, which is in line with previously characterized LIR motifs in *Drosophila* (Figure 1C). To verify the functionality of this putative LIR motif, we made point mutations of the Dwg LIR motif in positions 129 and 132 via alanine substitutions of the aromatic and hydrophobic residues (Y129A and I132A). This mutation significantly reduced its binding to Atg8a (Figure 1D). Furthermore, using ColabFold, we showed that Y129 and I132 dock nicely in hydrophobic pockets 1 and 2, respectively (Figure 2A), resembling the LIR motif of the known Atg8a-interacting protein Ref(2)P (Figure 2A) [7]. On the contrary, alanine substitutions at positions 129 and 132 disrupt the binding structurally (Figure 2B). These results show that the LIR motif at position 129–132 mediates the interaction between Dwg and Atg8a. Given the observed interaction between Dwg and Atg8a, we examined whether Dwg colocalizes with Atg8a. Immunofluorescence analysis showed that Dwg colocalizes with Atg8a in the nucleus upon fed conditions and in cytoplasmic puncta under starved conditions (Figure 2C,D). All together, these results indicate that Dwg is an Atg8a-interacting protein, and that this interaction is LIR motif-dependent.

### 2.2. Generation and Characterization of Dwg LIR^Y129A/I132A^ Mutant Flies

To elucidate the physiological significance of Atg8a-Dwg interaction in *Drosophila,* we used CRISPR to generate Dwg LIR^Y129A/I132A^ mutants (Appendix A). The Dwg LIR^Y129A/I132A^ mutants have two-point mutations (Y129A and I132A), which have been shown to interfere with the interaction to the LDS of Atg8a in GST pull-down assays. The Y129A/I132A mutation in mutant flies has been validated through genomic PCR and sequencing (Appendix A).

To examine whether the Y129A/I132A mutation has an effect on autophagy, we checked the lipidation of Atg8a by Western blotting [14]. We observed that Dwg LIR^Y129A/I132A^ mutants have significantly more Atg8a-II (lipidated form of Atg8a) compared to WT, which is an indication of increased autophagy (Figure 3A) [14]. As expected, we saw no Atg8a protein expression present in the Atg8a mutant samples compared to WT (Figure 3A). Ref(2)P, the *Drosophila* homolog of p62, marks ubiquitinated proteins for autophagic degradation and is commonly known to accumulate in autophagy mutants [7,10]. We have observed an accumulation of Ref(2)P in Atg8a mutant flies as expected, and minimal bands were seen in WT and Dwg LIR^Y129A/I132A^ flies (Figure 3B). We did not observe a significant difference in Ref(2)P expression between WT and Dwg LIR^Y129A/I132A^ flies, which suggests that (1) autophagy is not impaired in the Dwg LIR^Y129A/I132A^ mutants and (2) the increased autophagy observed in Dwg LIR^Y129A/I132A^ mutants is not sufficient to degrade more amounts of Ref(2)P.

To examine the physiological significance of increased autophagy in Dwg LIR^Y129A/I132A^ mutants, we applied oxidative and nutritional stress to them. We have found that the Y129A/I132A mutant flies show a higher resistance to starvation compared to WT flies (Figure 3C). The Dwg LIR^Y129A/I132A^ mutants have also shown a much higher resistance to oxidative stress compared to the control flies (Figure 3D), suggesting that the mutation provides a short-term advantage against stress-inducing factors. However, the lifespan of the Dwg LIR^Y129A/I132A^ flies was not increased compared to the WT flies (Figure 3E). The Dwg^8^ flies have shown a lower resistance to starvation and paraquat resistance compared to the WT and Dwg LIR^Y129A/I132A^ mutant flies and follow a similar pattern to the Atg8a flies in response to stress (Figure 3C,D). Atg8a flies are commonly used to study autophagy defects and mutations to the Atg8a gene are known to exhibit a significant reduction in lifespan [15]. Similar patterns were previously observed with Atg7, with mutations in the protein leading to shorter lifespans and increased sensitivity to oxidative and nutrient stress [16]. While Atg8a is involved in the functional aspects of autophagy and Atg7 is part of the autophagic machinery, they are both key proteins involved in autophagy and mutations in either one result in similar phenotypic presentation. Overall, it appears that the Dwg LIR^Y129A/I132A^ mutation has no long-term benefits in terms of survival but shows significant improvements in the resistance to oxidative stress and starvation in comparison to other genotypes. The Dwg^8^ mutant flies show opposite stress resistance trends in comparison to the Dwg LIR^Y129A/I132A^ mutant flies despite both lines having a mutation within the Dwg protein. This can be due to the nature of the mutation. The Dwg^8^ mutants contain a missense mutation (R14G) in the amino-terminal region of the protein that could disrupt protein–protein interactions or be related to Dwg’s transcriptional activity [12], whereas Dwg LIR^Y129A/I132A^ mutant flies have a mutation designed to specifically disrupt interaction with Atg8, which could be directly related to short-term advantages against stress conditions. While increased resistance to specific types of stress can contribute to better short-term survival under certain conditions (like oxidative stress and starvation shown in this study), there are various factors that can limit the overall impact on lifespan. An organism can often have multiple pathways for dealing with stress conditions. By enhancing a pathway that can help with oxidative stress and starvation, other compensatory mechanisms can be triggered that increase vulnerability in other areas, ultimately having a negative impact on overall lifespan despite improved resistance to specific stressors. This includes overactivation of stress response pathways like autophagy and antioxidant responses, which can lead to tissue damage and immune dysfunction, leading to limiting lifespan [17]. Ultimately, mutation-enhancing resistance to oxidative stress may not offer protection against other types of cellular and molecular damage related to aging [18].

While this manuscript was in preparation, another group found that Dwg interacts with Atg8a and is a transcriptional repressor of autophagy [8]. They were able to show that Dwg is targeted for degradation by autophagy, as evidenced by its interaction with Atg8a and degradation in response to autophagy-inducing treatments like rapamycin. Treatment with lysosomal inhibitor Bafilomycin was able to reverse the effects of rapamycin, indicating that Dwg is degraded by autophagy. Mutations to the Dwg LIR motif, which mediates the interaction with Atg8a, disrupted the interaction, leading to Dwg stabilization and inhibition of autophagy. Moreover, they have shown that Dwg translocates from the nucleus to the cytoplasm and is degraded in autophagosomes. Mutant Dwg proteins with impaired LIR motifs remain in the nucleus and inhibit autophagy, suggesting that Dwg-Atg8a interactions are crucial for regulating autophagy. They also used chromatin immunoprecipitation followed by next-generation sequencing (ChIP-seq) and they have also found that Dwg targets chromatin regions linked to autophagy-related genes [8]. ChIP-seq also confirmed that Dwg binds near insulator elements in four key ATG genes including Atg1, Atg3, Atg13, and Atg17, and suppresses their transcription, thereby acting as a negative regulator of autophagy [8]. This finding in combination with their qPCR results, which show upregulation of autophagy genes in Dwg mutant flies, suggests that Dwg interacts at the transcription level to repress autophagy [8]. Our results are consistent with these findings, confirming an LIR-dependent interaction with Atg8a and showing increased levels of Atg8a-II in Dwg mutant flies. This suggests increased autophagy when Dwg is unable to successfully negatively regulate key ATG genes via its interaction with Atg8a. Our findings also show that Dwg and Atg8a colocalize under fed and starved conditions, further confirming the physical association between the two proteins and supporting the idea that Dwg plays a direct role in the regulation of autophagy.

We have previously demonstrated that the transcription factor Sequoia interacts with Atg8a and plays a role in regulating the expression of autophagy-related genes and revealed a novel nuclear role for Atg8a, which is linked with the transcriptional regulation of autophagy genes [7]. Our findings in this paper show that another transcription factor, Dwg, has a similar role and contributes to the transcriptional regulation of autophagy, especially under stress conditions, and given the similarities between both transcription factors, it is possible that they may have complementary roles in autophagy regulation. Conclusively, our results provide important information on the transcriptional regulation of autophagy in *Drosophila* and contribute to a broader understanding of autophagy regulation at the transcriptional level.

## 3. Materials and Methods

### 3.1. Fly Husbandry and Generation of Transgenic Lines

Flies used in experiments were kept at 25 °C, 70% humidity, and raised on a cornmeal-based diet. The dwg^8^ (BDSC4094 dwg^8^/FM7a/Dp(1:3:Y)w^+^) and Atg8a mutant (BDSC14639 y [1] P{y[+mDint2] w[BR.E.BR] = SUPor- P}Atg8a[KG07569]/FM7c) flies were obtained from Bloomington *Drosophila* stock center.

The UAS-mCherry-Atg8a flies have been described previously by our lab [19]. Yeast two-hybrid screening was performed as described in [20]. Dwg LIR^Y127A/I132A^ CRISPR flies were created using site-directed mutagenesis and genome editing via CRISPR/Cas9 homology-dependent repair, using guide RNA and a dsDNA plasmid donor, performed by Wellgenetics Inc. (Taipei, Taiwan, China) [21]. gRNA sequences GCAGGCGTTCGACTCCACCG[AGG] and CATGATCGAGATGCTCAACG[AGG] were cloned into U6 promoter plasmid(s). The cassette PBacDsRed, which includes 3xP3-DsRed flanked by PiggyBac terminal repeats and two homology arms containing a point mutation, was cloned into pUC57-Kan to serve as the donor template for repair. Dwg CG2711-targeting gRNAs and hs-Cas9 were provided in DNA plasmids, along with the donor plasmid for microinjecting into the embryos of the control strain w [11,18]. F1 flies carrying the selection marker 3xP3-DsRed were then validated through genomic PCR and sequencing. The CRISPR mechanism introduced a break in the dwg/CG2711 gene, which was subsequently replaced by the PBacDsRed cassette. Finally, the PBacDsRed cassette was excised, effectively reconstituting the functional gene, which is now the Dwg LIR^Y127A/I132A^ mutant form, expressed across the entire genome of the selected fly strains. Mutant flies were further validated by genomic PCR and sequencing.

### 3.2. Protein Expression and Purification for GST Pull-Down Assay

Both the GST fusion bait and the His-labeled prey proteins were expressed in Escherichia coli BL21 Rosetta (DE3) (Novagen-Millipore, 70954) and incubated in liquid cultures. After reaching optimum density (0.6 Au at 600 nm wavelength), we induced protein construct expression with IPTG, at 0.5 mM final concentration. Cultures were left to further incubate at 20 °C for 16 h following IPTG induction. Bacteria were pelleted and re-suspended in lysis buffer (25 mM Tris pH 7.4 • 100 mM NaCl • 2 mM EDTA), additionally supplemented with 0.01% β-mercaptoethanol, and 1 μg/μL lysozyme (final concentrations). Cell integrity was disrupted by sonication using an EpiShear™ Probe Sonicator (in pulses 10 s ON, 5 s OFF, 30% amplitude) for 2 min per sample. Protein content was collected as the supernatant of the following centrifugation at 20,000 rpm, 4 °C for 20 min.

Both the GST bait and His prey lysates were incubated with Glutathione Sepharose^®^ 4B (GE Healthcare/Cytiva, 17-0756-01) for 30 min at 4 °C. Subsequent washes were carried out in high-salt (25 mM Tris pH 7.4 • 500 mM NaCl • 2 mM EDTA) and low-salt wash buffers (25 mM Tris pH 7.4 • 50 mM NaCl • 2 mM EDTA). After the washes, each pre-cleared His prey lysate was equally distributed to its respective GST bait-enriched beads and samples were co-incubated for 2 h at 4 °C. They were subsequently washed with 0.01% mercaptoethanol-supplemented lysis buffer and imidazole buffer (lysis buffer recipe + 10 mM imidazole). In preparation for gel loading, samples were finally re-suspended in equal volume of 2× Laemmli solution and denatured at 80 °C for 10 min, prior to SDS-PAGE and subsequent Western blot. For detection of all His prey proteins, we used an anti-6xHis antibody (Abcam, ab18184) at a 1:5000 dilution, while GST-tagged baits were visualized by Ponceau S stain used at a 0.2% working concentration.

### 3.3. Immunohistochemistry

Larva fat bodies were dissected in PBS and fixed for 30 min in 4% formaldehyde at room temperature. Blocking, as well as primary/secondary antibody incubations, was performed in PBT (0.3% BSA • 0.3% Triton-X100 in PBS). Primary and secondary antibodies were incubated overnight at 4 °C, or for 2 h at room temperature, in PBT.

Anti-Dwg (anti- Zw5, DSHB) primary was used at 1:5 dilution. Hoechst 33,342 DNA staining dye (New England Bioloabs #4082, 1:1000 in PBS) was used to visualize nuclei. Washes were performed in PBW (0.1% Tween-20 in PBS).

All images were acquired in Carl Zeiss LSM710 or LSM880 confocal microscopes, using a 63× Apochromat objective lens.

### 3.4. Analysis Using ColabFold/PyMOL

Sequences of proteins analyzed were obtained from UniProt. ColabFold2 version1.5.5 (available at: https://colab.research.google.com/github/sokrypton/ColabFold/blob/main/AlphaFold2.ipynb accessed on 2 November 2022) was used for protein structure and docking prediction. PyMOL (Molecular Graphics System, Version 2.3.5 Schrödinger, LLC (New York, NY, USA)) was used to analyze and visualize the predicted protein structures obtained from ColabFold2 using the align function. The Dwg LIR^Y129A/I132A^ mutation was manually added to the Dwg FASTA sequence, for alignment comparison. Docking structure images were copied into Microsoft PowerPoint for processing and labeling.

### 3.5. Protein Extraction and Western Blotting

Protein content was extracted from the full fly body in RIPA lysis buffer (50 mM Tris pH 7.4, 150 mM NaCl, 1% Igepal, 0.5% sodium deoxycholate, 0.1% SDS supplemented with cOmpleteTM ULTRA EDTA-free protease inhibitor cocktail (Roche (Basel, Switzerland), #5892791001)) using a motorized mortar and pestle. Protein concentrations were determined by the Bradford method.

The 100 μg protein samples were loaded on 18% polyacrylamide gels and run at 75 V for 4 h. The gels were then transferred onto PVDF membranes (cold wet transfer in 10% ethanol O/N at 30V followed by 30 min at 100 V). Membranes were blocked in 5% BSA in TBST (0.1% Tween-20 in TBS) for 1 h. Primary against GABARAP (ab109364) was diluted 1:1000 in TBST and incubated overnight at 4 °C with gentle agitation. HRP-coupled secondary antibodies binding was performed at room temperature (RT) for 1 h in 1% BSA dissolved in TBST and ECL mix incubation for 2 min. All washes were performed for 10 min in TBST at RT.

### 3.6. Lifespan Assays

For starvation assays, three-to-five-day-old female flies were collected and separated into groups of 15 per vial. A piece of filter paper soaked in sterile water was placed inside the vials with additional liquid added when necessary. Flies were kept at 25 °C for the duration of the experiment and deaths were recorded daily.

For lifespan analysis, newly eclosed female flies were separated into vials in cohorts of 10–15 and kept at 25 °C for the duration of the experiment. Flies were transferred to fresh food every 2–3 days and deaths were logged until the final death was recorded.

For oxidative stress experiments, paraquat was mixed with the food for a final concentration of 30mM and dead flies were counted and recorded daily.

### 3.7. Statistics and Figures

All statistical analyses were performed with GraphPad Prism 10 software (version 10.4.0). Fold change was calculated by normalizing to loading control and compared to the protein of interest. Figures were assembled in Adobe Illustrator 2023 (version 27.9.6).

## Figures and Tables

**Figure 1 cells-13-01897-f001:**
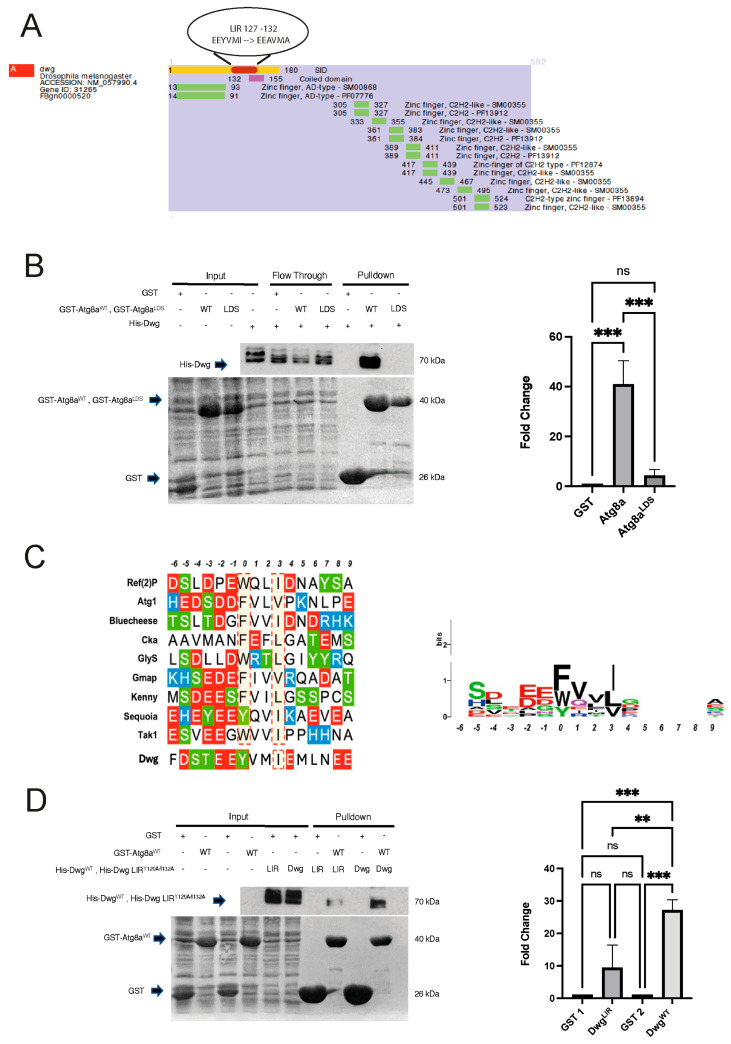
(**A**) Yeast two-hybrid data showing interaction of the Selected Interaction Domain fragment (SID) in yellow in the region between 1 and 180 of Dwg with Atg8a. Using the online iLIR software, an LIR motif with the sequence EEYVMI was identified within the SID region at position 127–132. (**B**) GST pull-down assay confirming interaction between Dwg and Atg8a^WT^ and showing a reduced interaction between Dwg and Atg8a^LDS^ mutant. Experiment was performed in triplicate and data were analyzed by one-way ANOVA with a post hoc Tukey test, *** *p* < 0.001, ns: not significant based on the statistical analysis performed. (**C**) LIR motif consensus sequences of known *Drosophila* LIR motif-containing proteins (LIRCPs). (**Left** panel) Alignment of the experimentally verified functional LIR motifs identified in *Drosophila* LIRCPs. The LIR peptide extends six amino acids N-term and nine amino acids C-term from the Atg8a LDS HP1-binding residue (designated as position “0”). Highlighted red dotted boxes denote the critical amino acids at positions “X0” and “X3” of the LIR core that dock to the HP1 and HP2 sites, respectively, within the Atg8a LDS pocket. Acidic amino acids are represented with red, basic amino acids with blue, and with green are serine and threonine residues that may be targets of de/phosphorylation cycles. (**Right** panel) Sequence logo map for the conservation of each residue with respect to its position in the extended LIR motif. Sequence logo graph constructed using WebLogo (available at: https://weblogo.berkeley.edu/). (**D**) GST pull-down assay looking at the interaction of Dwg with Atg8a when a mutation has been introduced in the predicted LIR motif responsible for the Atg8a binding. Alanine substitutions were introduced in the predicted binding sequence at positions 129 and 132 (EEYVMI EEAVMA). Data were analyzed by one-way ANOVA with a post hoc Tukey test, *n* = 3, *** *p* < 0.001, ** *p* < 0.01, ns: not significant based on the statistical analysis performed.

**Figure 2 cells-13-01897-f002:**
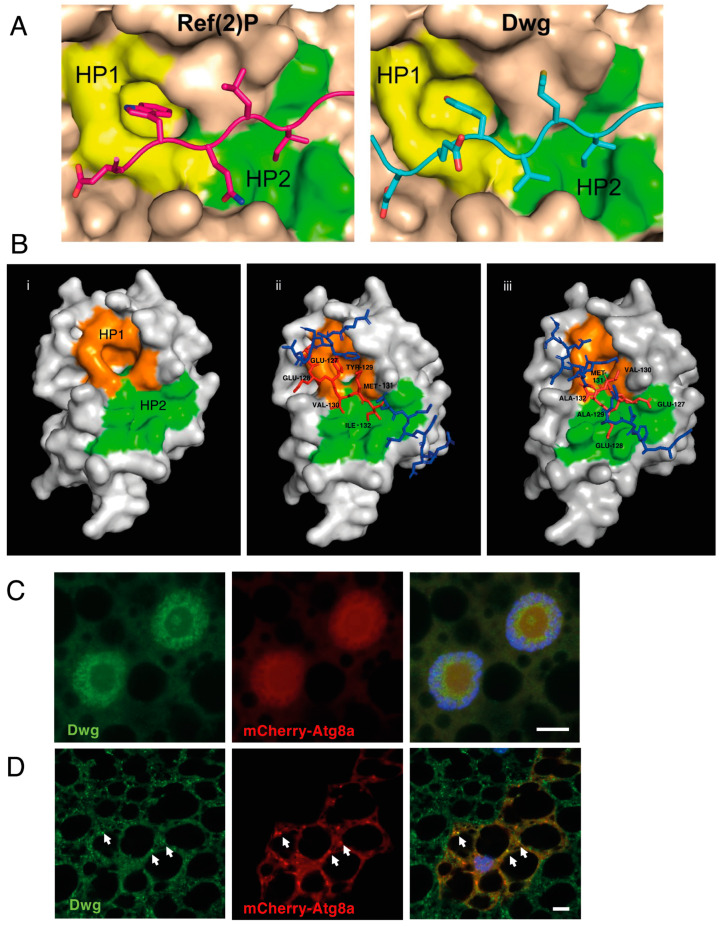
(**A**) Superimposition of Ref(2)P (**left**) and Dwg (**right**) protein backbones docking to the HP1 (yellow) and HP2 (green) sites of Atg8a. (**B**) (**i**) Three-dimensional superimposition of Atg8a HP1 (orange) and HP2 (green) sites. (**ii**) Dwg protein backbone (blue) and the predicted LIR motif (red) shown docking to the LDS of Atg8a. Tyrosine seen docking in the HP1 and isoleucine docking to the HP2 pockets. (**iii**) Dwg protein backbone (blue) and the predicted LIR motif with two alanine mutations introduced to the sequence. The models show a lost interaction and lack of docking to the HP1 and HP2 sites. (**C**) Confocal micrographs of larval fat bodies under fed conditions expressing mCherry-Atg8a and stained for endogenous Dwg (green). Scale bar: 10 microns. (**D**) Confocal micrographs of larval fat bodies under starved conditions (4 h, 20% sucrose) expressing mCherry-Atg8a and stained for endogenous Dwg (green). Arrows show colocalization between cytoplasmic Dwg puncta and the autophagic marker mCherry-Atg8a. Scale bar: 10 microns.

**Figure 3 cells-13-01897-f003:**
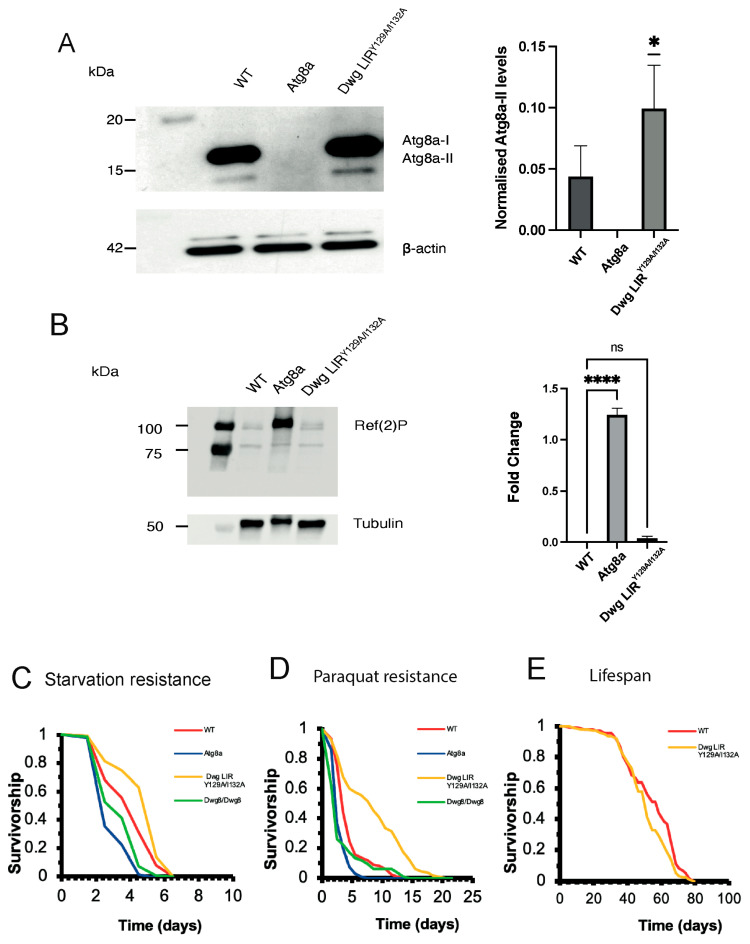
(**A**) Atg8a lipidation Western blot for Atg8a-I and Atg8a-II using GABARAP in 3-week-old adult female flies. Quantification for normalized levels of Atg8a-II in Dwg LIR^Y129A/I132A^ mutant flies. Actin used as loading control. *n* = 3, * = *p* ≤ 0.1. One sample T test and Wilcoxon post hoc test conducted. Error bars represent SD. (**B**) Western blot of the same samples showing Ref(2)P levels in WT, Atg8a, and Dwg LIR^Y129A/I132A^ mutant flies. Data were analyzed by one-way ANOVA with a post hoc Tukey test, *n* = 3, **** *p* < 0.0001, ns: not significant based on the statistical analysis performed. (**C**) Dwg LIR^Y129A/I132A^ mutant flies show a higher resistance to oxidative stress. Survival of ~125 female flies of each genotype with 30 mM paraquat-supplemented food. Survival graph showing WT (red), Dwg LIR^Y129A/I132A^ (yellow), Dwg^8^/Dwg^8^(green), and Atg8a (blue). Log-rank statistical analysis was performed. WT vs. Atg8a **** *p* < 0.0001, WT vs. Dwg^8^/Dwg^8^ **** *p* < 0.0001, WT vs. Dwg LIR^Y129A/I132A^ **** *p* < 0.0001. (**D**) Dwg LIR^Y129A/I132A^ mutant flies show a higher resistance to starvation. Survival of ~125 female flies of each genotype on water-only diet. Log-rank statistical analysis was performed. WT vs. Atg8a **** *p* < 0.0001, WT vs. Dwg^8^/Dwg^8^ **** *p* < 0.0001, WT vs. Dwg LIR^Y129A/I132A^ **** *p* < 0.0001. (**E**) Dwg LIR^Y129A/I132A^ mutant flies show no change in lifespan compared to WT. Survival of ~150 female flies of each genotype was tracked over time and analyzed using a log-rank test. Female survival graph showing WT (red), and Dwg LIR^Y129A/I132A^ (yellow). WT vs. Dwg LIR^Y129A/I132A^ * *p* < 0.5 with a significance value of 0.013.

## Data Availability

The original contributions presented in the study are included in the article/Appendix A, further inquiries can be directed to the corresponding author.

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
