# Peer review of "Transcription Factor Deformed Wings Is an Atg8a-Interacting Protein That Regulates Autophagy"

_cells, 2024, doi:10.3390/cells13221897_

Round 1
Reviewer 1 Report
Comments and Suggestions for Authors
In the article titled “Transcription factor deformed wings is an Atg8a-interacting protein that regulates autophagy “the authors identified Dwg as a new Atg8a-interacting protein. I would like to acknowledge the quality and clarity of the presented work. The manuscript is well-structured, and the introduction thoroughly covers the essential background information. I recommend the publication of this study after the authors address the following key points:
-Include the p62/SQSTM1 expression level in figure 3.
-Furthermore, all figures should be presented in a higher resolution.
Reviewer 2 Report
Comments and Suggestions for Authors
This study gives new findings of the transcriptional regulation of autophagy, might be very interested to readers of this research area. Dwg is presented in different forms in the text, such as DWG, dwg, Dwg, dwg8, dwgY127A, it would be better in one consistent form. Some typing mistakes in the legend and in text.
Reviewer 3 Report
Comments and Suggestions for Authors
This brief report describes a nuclear transcription factor, Dwg, directly interacting with Atg8a in Drosophila. The interaction depends on the LIR motif. Dwg mutant flies have a higher survival rate under starvation and oxidative stress conditions. Unfortunately, most of their findings were reported by others in April 2023, and I don't see the point of publishing this manuscript one year later.
Overall, the data are poorly presented. Figure legends are not clear and figures are badly labeled. For instance, the amino acid labels in Fig. 2B (ii) and (iii) are hard to see even after zooming in. Fig.1A lists many Zinc Finger proteins or domains without any explanation. Fig. 1B, 1D, fold change is not defined in terms of how it is done. Fig. 3A shows Atg8a (mutant fly?) without mentioning in the result context. Fig. 3B. shows Atg8a, Dwg8 mutant flies but no discussion at all in the result section. Dwg8 and Atg8a flies are not described in the Method section.
Dwg LIRY129A/I132A should behave like Dwg8 mutant flies but they exhibit opposite survival phenotypes. While the authors only detected very minor ATG8a lipidation in the Dwg LIRY129A/I132A fly, a Nature Communication (2023) paper reported much stronger Atg8a lipidation in the Dwg8 mutant fly by Tang et al.
It is also not acceptable not to mention Tang et al's paper in the Introduction since it was published 18 months ago.
Round 2
Reviewer 1 Report
Comments and Suggestions for Authors
I accept in present form
Reviewer 3 Report
Comments and Suggestions for Authors
The revised manuscript addressed most of my concerns. Although the authors now cite Tang et al Nature Communication paper (2023) in the Introduction, they omit relevant overlap with the work shown here.
line 125: typo: "Similar patters" should be "similar patterns"
line 139: "," is missing before "which"